# The Impact of Inflammatory Stimuli on Xylosyltransferase-I Regulation in Primary Human Dermal Fibroblasts

**DOI:** 10.3390/biomedicines10061451

**Published:** 2022-06-19

**Authors:** Thanh-Diep Ly, Christopher Lindenkamp, Eva Kara, Vanessa Schmidt, Anika Kleine, Bastian Fischer, Doris Hendig, Cornelius Knabbe, Isabel Faust-Hinse

**Affiliations:** Institut für Laboratoriums-und Transfusionsmedizin, Herz-und Diabeteszentrum NRW, Universitätsklinik der Ruhr-Universität Bochum, Georgstraße 11, 32545 Bad Oeynhausen, Germany; tly@hdz-nrw.de (T.-D.L.); clindenkamp@hdz-nrw.de (C.L.); ekara@hdz-nrw.de (E.K.); vschmidt@hdz-nrw.de (V.S.); ankleine@hdz-nrw.de (A.K.); bfischer@hdz-nrw.de (B.F.); dhendig@hdz-nrw.de (D.H.); cknabbe@hdz-nrw.de (C.K.)

**Keywords:** ATP, caspase, cathepsin, fibroblast, fibrosis, inflammasome, inflammation, LPS, PXE, xylosyltransferase

## Abstract

Inflammation plays a vital role in regulating fibrotic processes. Beside their classical role in extracellular matrix synthesis and remodeling, fibroblasts act as immune sentinel cells participating in regulating immune responses. The human xylosyltransferase-I (XT-I) catalyzes the initial step in proteoglycan biosynthesis and was shown to be upregulated in normal human dermal fibroblasts (NHDF) under fibrotic conditions. Regarding inflammation, the regulation of XT-I remains elusive. This study aims to investigate the effect of lipopolysaccharide (LPS), a prototypical pathogen-associated molecular pattern, and the damage-associated molecular pattern adenosine triphosphate (ATP) on the expression of *XYLT1* and XT-I activity of NHDF. We used an in vitro cell culture model and mimicked the inflammatory tissue environment by exogenous LPS and ATP supplementation. Combining gene expression analyses, enzyme activity assays, and targeted gene silencing, we found a hitherto unknown mechanism involving the inflammasome pathway components cathepsin B (CTSB) and caspase-1 in XT-I regulation. The suppressive role of CTSB on the expression of *XYLT1* was further validated by the quantification of *CTSB* expression in fibroblasts from patients with the inflammation-associated disease Pseudoxanthoma elasticum. Altogether, this study further improves the mechanistic understanding of inflammatory XT-I regulation and provides evidence for fibroblast-targeted therapies in inflammatory diseases.

## 1. Introduction

Inflammation is the physiological response to tissue damage caused by harmful stimuli, such as pathogens; damaged cells releasing endogenous antigens and alarmins, such as adenosine triphosphate (ATP); or irritants [1]. The inflammation appears within minutes of the initial trauma and is tightly controlled to maintain tissue homeostasis. Uncontrolled inflammatory responses or inefficient inflammation resolution may contribute to the emergence of inflammatory and autoimmune diseases. Healing of the injured tissue during the dissolving of inflammation must be strictly balanced as excessive tissue remodeling can lead to fibrosis and scarring of the tissue involved. The tissue microenvironment controls the behavior of local immune cells in chronic infection and inflammation. Tissue-resident fibroblasts not only play a key role in extracellular matrix (ECM) synthesis, and the remodeling and maintenance of tissue homeostasis, they also contribute to the activation and modulation of immune responses by acting as immune sentinel cells upon the detection of pathological stimuli [2]. Lipopolysaccharide (LPS), an endotoxin from Gram-negative bacteria, is a prototypical pathogen-associated molecular pattern and a potent mediator of sepsis and septic shock. It exerts its main effect by the activation of cell surface toll-like receptors, which are predominant pattern-recognition receptors expressed on immune and nonimmune cells. Human fibroblasts regulate the inflammatory response via the toll-like receptor 4 activated by LPS, which led to the nuclear factor kappa-light-chain-enhancer of activated B cells (NF-κB) transcription factor activation, and cytokine and chemokine expression, including interleukin (IL)-1β and IL-8, resulting in immune cell recruitment and local tissue inflammation [3,4]. The IL-1β is induced by LPS in human fibroblasts with major roles in inflammation, innate immune response and fibrosis [5,6]. Excess production of IL-1β can be detrimental to the host; therefore, its production is closely controlled at various levels to assure its proper release. One of the latter is the NF-κB-dependent pro-IL-1β synthesis, and the other is the caspase-1 (CASP1)-dependent proteolytic conversion of pro-IL-1β to mature IL-1β by CASP1-containing multi-protein complexes, termed inflammasomes, that are activated by cellular infections or stress. Inflammasomes are not only restricted to classical immune cells, including macrophages or T cells; they are also activated in a variety of different cell types, such as epithelial cells and fibroblasts [7,8,9]. Mammalian cells export a substantial majority of their proteins, including IL-8, via the endoplasmic reticulum and Golgi-dependent pathway. The IL-8 protein secretion appears to be directly related to cellular levels of *IL8* mRNA [10]. Some proteins, however, are secreted via unconventional mechanisms. Among the latter is the proinflammatory cytokine IL-1β. CASP1 (EC 3.4.22.36) is a cysteine protease and regulator of inflammatory responses through its capacity to process inflammatory cytokines and their involvement in the secretion of those leaderless proteins. Cathepsins are members of the papain family of cysteine proteases. They are found primarily in endosomes and lysosomes and are important for protein breakdown, as well as in the cytoplasm, the cell nucleus and the extracellular space [11]. Their extra-lysosomal localization and activity are often associated with the pathogenesis of cancer, neurodegeneration and metabolic diseases [12]. Extracellular cathepsins participate in ECM remodeling by degrading structural components, such as collagens, elastin or proteoglycans (PGs) [13,14]. Cathepsins are also involved in pro-inflammatory cytokine processing. Cathepsin B (CTSB; EC 3.4.22.1) plays a critical role in inflammatory responses and IL-1β processing in immune and nonimmune cells. It is essential for NF-κB and CASP1 activation [7,13,15,16,17]. The secretion of cysteine cathepsins is related to their overexpression due to extracellular stimuli by different cytokines. Increased levels of CTSB have been detected in synovial fluids of patients with rheumatoid arthritis and osteoarthritis, implying their participation in inflammation and cartilage destruction [18].

Proteoglycans are crucial components of the ECM and are important for cell adhesion, migration, signal transduction and immune response. They are composed of a PG core protein and glycosaminoglycan (GAG) chains. The human xylosyltransferase (XT; EC 2.4.2.26) is a key enzyme in GAG biosynthesis. It catalyzes the transfer of xylose from sugar-donor UDP-xylose to specified serine residues in the PG core protein as an acceptor. Despite its intracellular localization, the enzyme is cleaved from the membrane of the Golgi apparatus by an unknown mechanism requiring cathepsin activity and is subsequently secreted into the extracellular space. Therefore, extracellular XT activity can be quantified in different samples, including human serum or cell culture supernatants, as an indicator of the actual PG synthesis rate. The determination of XT activity in the dermal fibroblast culture over a certain period revealed an accumulation of XT activity in the cell culture supernatant, whereas the intracellular amount of XT remained constant over time. Thus, two regulatory mechanisms are assumed to control the cellular XT activity: one at the transcriptional level affecting the relative availability of transcripts and another operating post-transcriptionally, modulating the protein secretion. In humans, there are two distinctive XT isoforms, XT-I and XT-II, encoded by the genes *XYLT1* and *XYLT2*, respectively [19]. The expression of the *XYLT1* and *XYLT2* isoforms is regulated differently by exogenous cytokines and growth factors, such as profibrotic transforming growth factor beta 1 (TGF-β1) and activin A. While the increase in *XYLT1* mRNA expression and XT-I activity is a marker for myofibroblast differentiation and correlates with an aberrant PG biosynthesis in fibrotic conditions [5,20,21,22], the *XYLT2* mRNA expression is often unaffected by exogenous stimuli and resembles the expression of a robust housekeeping gene. Aberrant or suppressed *XYLT1* mRNA expression and XT activity have been observed in catabolic PG-associated joint diseases, such as rheumatoid arthritis and osteoarthritis. The *XYLT1* expression is downregulated by the catabolic cytokine IL-1β in human primary chondrocytes and cartilage explants [23], while the relative *XYLT1* mRNA expression of primary normal human dermal fibroblasts (NHDF) is unchanged upon IL-1β stimulation [5]. Therefore, cell type-specific differences in cytokine-mediated XT-I regulation exist. The IL-1β and inflammasome pathway was reported to play an important role in chronic liver inflammation leading to fibrosis and cirrhosis [24]. Although an increase in serum XT activity was found in patients with liver fibrosis, the inflammatory XT-I regulation in fibroblasts, and the immune sentinel cells, during the inflammatory tissue remodeling stage remain elusive.

It remains unknown if fibrosis can occur without inflammation or not in the final stage of diseases. One example of inflammation-enhancing fibrogenic responses is the pro-inflammatory cytokine IL-1β that was shown to positively enhance the TGF-β signaling pathway by inducing the expression of TGF-β receptor type II in NHDF that lies upstream of the canonical Smad protein-mediated signaling cascade [5]. On the contrary, inflammation has been found in several fibrotic tissues with no inflammatory features [1,25]. Thus, an inflammatory response subsequent to major tissue damage may not always lead to fibrosis and may actually inhibit fibrosis and vice versa [26]. These assumptions were strengthened by the fact that TGF-β possesses an anti-inflammatory and immunosuppressive role in regulating inflammatory and adaptive immune responses [27].

Pseudoxanthoma elasticum (PXE) is a metabolic inherited disorder marked by elastic fiber fragmentation and ectopic calcification that affects the skin, the eye and the vascular system. It is caused by a deficiency of ABCC6, a member of the ATP-binding cassette superfamily. Cellular and molecular biomarkers indicate premature aging in PXE [28]. Whole sequence analysis identified four modifier genes in PXE that belong to the IL1B and inflammasome signaling pathway [29]. Furthermore, PXE has several similarities with the prominent premature aging disorder Hutchinson–Gilford progeria syndrome, which has also been shown previously to involve the aberrant regulation of inflammasome pathway components [30]. In addition, there is growing evidence indicating a correlation between ABCC6 dysfunction and dyslipidemia. Primary cells derived from patients with PXE provide a useful disease model for inflammatory conditions because inflammasome pathway components link inflammation and cholesterol trafficking to premature cellular senescence [12,28,31,32,33].

The identification of disease-specific alterations in fibroblasts is the crucial step in unraveling the molecular pathways’ underlying pathological changes. This study was performed to gain new insight into the inflammatory response of fibroblasts as key immune sentinel cells and addresses the gene expression regulation of the fibrotic marker *XYLT1* and XT-I activity under inflammatory conditions. In addition to the cell culture models with NHDF at low and high cell densities, an inflammation-associated disease model with fibroblasts derived from patients with PXE was used to verify the initial results obtained in this study.

## 2. Materials and Methods

### 2.1. Materials and Reagent Preparation

Highly purified, γ-irradiated LPS from *E. coli* serotype O111:B4 was obtained from Sigma-Aldrich (St. Louis, MO, USA) and diluted in water prior to usage. Cell-culture-tested ATP disodium salt was purchased from Sigma-Aldrich (St. Louis, MO, USA). A mildly acidic aqueous ATP stock solution (200 mM, pH 3.5) was prepared in water and neutralized with 1 M NaOH (Sigma-Aldrich, St. Louis, MO, USA) prior to cell culture usage. The exact pH of the ATP solution (100 mM ATP, pH 7.5) acquired was determined by absorption spectroscopy and calculated using the Lambert–Beer law. The siRNAs, the transfection reagents, and the Opti-MEM I medium were acquired from Thermo Fisher Scientific (Waltham, MA, USA).

### 2.2. Primary Cell Culture

Adult NHDF were obtained from Coriell (Camden, NJ, USA). Dermal fibroblasts from patients with PXE were provided according to the authors’ description in [28], who also listed the clinical characteristics of the PXE patients. The study was approved by the Ethics Committee of the HDZ NRW, Department of Medicine, Ruhr University of Bochum (registry no. 32/2008, approval date is 3 November 2008). Primary cells were maintained under standardized conditions (37 °C, 5% CO_2_) as a monolayer culture in tissue culture dishes (100 × 20 mm, Greiner bio-one, Frickenhausen, Germany) with Dulbecco’s modified Eagle’s medium without phenol-red addition (DMEM; Thermo Fisher Scientific, Waltham, MA, USA). DMEM was supplemented with either 10% (*v*/*v*) fetal calf serum (FCS; Biowest, Nuaillé, France) or 10% (*v*/*v*) lipoprotein-deficient FCS (LPDS) and 4 mM L-glutamine (PAN Biotech, Aidenbach, Germany), 1% (*v*/*v*) Penicillin-Streptomycin-Amphotericin B solution (100×; PAN Biotech, Aidenbach, Germany), as described previously [34]. The LPDS was prepared according to our previous work [28]. Medium changes were performed twice a week. The subculturing of near confluent primary NHDF was performed with an expansion ratio of 1:3 utilizing 0.05% (*v/v*) trypsin (PAN Biotech, Aidenbach, Germany) in Dulbecco’s phosphate buffered saline (PBS, 1×; Thermo Fisher Scientific, Waltham, MA, USA).

### 2.3. Cell Treatment and Sample Preparation

Two cell culture models established earlier were utilized to study the effect of LPS on NHDF [21,28]. The effect of LPS on proto-myofibroblasts was analyzed in low-density primary fibroblast cultures with 50 cells per mm^2^, while the effect of LPS on inactivated fibroblasts was investigated using the high-density culture model with 177 fibroblasts per mm^2^. Irrespective of the cell culture model used, cells were cultured in fully supplemented growth medium with FCS for 24 h before cell treatment. Treatments were carried out with a final LPS concentration of 0.1 μg/mL or 1.0 μg/mL, and/or a final ATP concentration of 5 mM diluted in fully supplemented growth medium for the time points indicated. At every sampling time, negative controls, treated with solvent or vehicle only, were included.

A total of 2.9 × 10^5^ cells per dish (60 × 50 mm; Corning Inc., Corning, NY, USA) were used for siRNA knockdown procedures and maintained in antibiotic-free Opti-MEM I medium supplemented with FCS. Reverse transfection was carried out with Lipofectamine 2000 reagent. The transfection mixture contained a silencer predesigned siRNAs targeting *CTSB* or *CASP1* or contained a non-targeting siRNA control diluted in Opti-MEM I medium at a concentration of 50 nM siRNA per well. The transfection mixture was replaced after 24 h with fully supplemented DMEM for another 24 h. Transfected cells were maintained in growth medium supplemented with 0.1 μg/mL LPS for 24 h until subsequent lysis.

The cell monolayer was washed with 1× PBS and incubated with 0.35 mL RA1-buffer (Macherey-Nagel, Düren, Germany) for cell lysis and sample preparation for analysis via quantitative real-time PCR (qRT-PCR). The cell culture supernatant was collected after 24 or 48 h to analyze the extracellular XT-I activity. The corresponding cell monolayer was incubated with 0.75 mL Nonidet P 40-lysis buffer and subsequently prepared as formerly described [34] for the analysis of the intracellular XT-I activity. The intracellular fraction was also used for protein quantification via bicinchonic acid (BCA) assay. All experiments were carried out in biological and technical triplicates per number *n* of donor-derived primary cell cultures unless otherwise stated.

### 2.4. Cell Proliferation Assay

The cell proliferation and viability to exogenous stimuli were spectrometrically quantified by using WST-1 reagent (Roche, Basel, Switzerland) according to the manufacturer’s instructions. A total cell number of 1700 per cavity of a 96-well culture plate (Greiner bio-one, Frickenhausen, Germany) were used and maintained in fully supplemented growth medium for 24 h before treatment with LPS and ATP. After 20 h, the WST-1 reagent was added to each well. The absorbance at the wavelength 440 nm and 590 nm as a reference were determined at time points 0, 1, 2, 3, and 4 h after WST-1 supplementation using a multiplate reader (Tecan, Männedorf, Switzerland).

### 2.5. BCA Assay

The BCA protein assay was performed to determine the protein concentration of a lysate sample with detergent supplementation [35]. The assay was conducted in a format according to the manufacturer’s instructions of the Pierce BCA Protein Assay Kit (Thermo Fisher Scientific, Waltham, MA, USA). In brief, a protein standard curve involving six bovine serum albumin (Sigma-Aldrich, St. Louis, MO, USA) standards ranging from 0 to 1000 g/L was prepared with Nonidet P 40-lysis buffer as solvent. The BCA working solution consists of 50 parts of a BCA solution and 1 part of a Cu^2+^ solution. A total of 200 μL BCA working solution was added to 25 μL standard or protein solution per well of a 96-microplate and incubated for 30 min at 37 °C. Thereafter, the absorbance at 562 nm was measured. The absorbances of the protein standards were used to determine the protein concentrations of the unknown samples in respect of their absorbance values by linear regression.

### 2.6. XT-I Activity Determination by Mass Spectrometry

The selective determination of a sample’s XT-I activity was performed by ultra-performance liquid chromatography/electrospray ionization tandem mass spectrometry (UPLC-ESI-MS/MS). The assay is based on the XT-I catalyzed transfer of xylose on a XT-I selective acceptor peptide after a fixed reaction time. The in-silico determination of the XT-I activity from cell culture supernatants (extracellular XT-I) and cell lysates (intracellular XT-I) was conducted, as described previously [5]. The quantified XT-I activity is expressed in arbitrary units (AU) and normalized to the total protein content of the respective lysate sample.

### 2.7. RNA Extraction and cDNA Synthesis

The extraction of RNA from whole cell lysates and the subsequent cDNA synthesis for qRT-PCR analysis were conducted, as described previously [34].

### 2.8. QRT-PCR Analysis

The gene expression analysis via qRT-PCR was conducted as described in our previous work using a SYBR green-based amplicon detection [34]. The primer sequences utilized are listed in Appendix A and in [34]. The geometrical mean of the expression levels of three reference genes (*SDHA*, *RPL13A* and *B2M)* were used for expression normalization. The relative target gene expressions of two samples were calculated on the basis of the ΔΔC_T_ method, which considered the PCR efficiency of the primer systems used [36]. For the relative comparison of multiple biological samples per experiment, all normalized mRNA expressions were referred to the expression of the target gene of one donor-derived primary cell sample.

### 2.9. Statistical Analysis

The data values shown are means ± standard error of the mean (SEM). Because of the lack of Gaussian distribution (Shapiro–Wilk normality test), the nonparametric two-tailed Mann–Whitney *U* test was utilized for data analysis. All analyses were conducted with the software GraphPad Prism 9 (GraphPad Software version 9.1.1, La Jolla, CA, USA). A probability *p* value of less than 0.05 was considered to be statistically significant.

## 3. Results

### 3.1. Time- and Concentration-Dependent Decrease of XYLT1 mRNA-Expression by LPS in Primary Skin Fibroblasts

The impact of LPS on the pro-inflammatory and fibrotic gene expression profile of NHDF was evaluated using a cell culture model established by the authors of [21], which is based on a low cell density culture condition on hard tissue culture substrates promoting the in vitro generation of protomyofibroblasts. The NHDF were treated without or with different concentrations of LPS for 24 h (Figure 1) or 48 h (Appendix A) to choose the most suitable LPS concentration and treatment duration for the subsequent analyses. The relative mRNA expression of myofibroblast marker *XYLT1* and its non-fibrosis-related isoform *XYLT2* were analyzed by qRT-PCR after both time points. The expression of the known LPS-inducible inflammatory mediator gene *IL1B* [7,8,37], which has previously been shown not to regulate the *XYLT1* mRNA expression of NHDF after a treatment period of 48 h [5], was used to control the efficacy of the LPS treatment applied.

The cell treatment with 0.1 μg/mL LPS decreased the relative *XYLT1* mRNA expression significantly (0.6 ± 0.1-fold, *p* < 0.001), not effecting the relative *XYLT2* mRNA expression of the cells. The usage of 1.0 μg/mL LPS diminished the relative *XYLT1* and *XYLT2* expression significantly (both 0.7 ± 0.1-fold, *p* < 0.001) compared to the untreated controls after a culture period of 24 h (Figure 1A,B). The treatment of NHDF with both LPS concentrations of 0.1 or 1.0 μg/mL for 24 h led to a significant increase in the respective *IL1B* mRNA expression by 140 ± 40-fold or 138 ± 36-fold (*p* < 0.0001), respectively (Figure 1C). When we extended the LPS cell treatment period from 24 to 48 h, we observed a slight 0.7 ± 0.2-fold (*p* < 0.05) decrease in the relative *XYLT1* mRNA of 1.0 μg/mL NHDF treated with LPS and a 0.8 ± 0.1-fold (*p* < 0.05) decreased *XYLT2* mRNA expression of 0.1 μg/mL NHDF treated with LPS, compared to the respective controls (Appendix A). Furthermore, the treatment of NHDF with both LPS concentrations of 0.1 or 1.0 μg/mL for 48 h led to a significant 232 ± 28-fold or 296 ± 41-fold (*p* < 0.0001) increase, respectively, in the relative *IL1B* mRNA expression of NHDF treated with LPS (Appendix A). We conclude from the latter that the LPS treatment worked properly in our cell culture system. Furthermore, the results showed that LPS has a suppressive effect on the fibrosis-related *XYLT1* mRNA expression of NHDF.

Since the cell density has been shown to influence the relative *XYLT1* mRNA expression of NHDF [21], we next analyzed the impact of the cell density used on the relative *XYLT1* mRNA expression of NHDF treated with LPS. We chose a cultivation period of 24 h for the LPS treatment of NHDF and increased the cell density from 50 to 177 cells/mm^2^. The higher cell density did not promote myofibroblast differentiation [34], thereby retaining the native phenotype of the NHDF cultured. When analyzing the relative changes in the mRNA expression of *XYLT1*, *XYLT2* and *IL1B* upon the LPS treatment of high-density cultured NHDF (Figure 2), we came across a similar transcription pattern to those of our previous experimental setup.

We found a significant *XYLT1* mRNA expression decrease (0.40 ± 0.1-fold, *p* < 0.01) in NHDF treated with LPS compared to untreated controls by using a concentration of 0.1 μg/mL LPS and a treatment period of 24 h. No significant changes in the relative *XYLT1* mRNA expression were found upon the treatment of high-cell-density-cultured NHDF with 1.0 μg/mL LPS for 24 h (Figure 2A). In addition, the LPS concentrations used did not alter the relative *XYLT2* mRNA expression of treated NHDF compared to controls (Figure 2B). Compared to untreated controls, the mRNA expression of the LPS-inducible control gene *IL1B* was 361 ± 57-fold and 385 ± 56-fold (*p* < 0.0001) increased in NHDF upon LPS incubation with 0.1 and 1.0 μg/mL LPS for 24 h, respectively (Figure 2C). It can be concluded that LPS is a potent suppressor of the relative *XYLT1* mRNA expression in both low- and high-cell-density-cultured NHDF. Interestingly, the high-cell-density culture conditions showed a more pronounced *XYLT1* mRNA expression LPS-mediated decrease in NHDF, not affecting the relative *XYLT2* mRNA expression of the cells. The LPS was also shown to be a potent inducer of inflammatory cytokine *IL1B* mRNA expression in our high-cell-density-culture system.

Fibroblasts contribute to the initiation of the inflammatory response by attracting immune cells; therefore, we quantified the relative expression of a representative pro-inflammatory chemokine CXCL8 (*IL8*) that had previously been shown to be increased in NHDF upon LPS treatment [38]. We observed a significant LPS-mediated *IL8* mRNA expression increase by 280 ± 54-fold and 254 ± 43-fold (*p* < 0.0001) in 0.1 and 1.0 μg/mL NHDF treated with LPS, respectively (Figure A1A). The increased cytokine expression observed in fibroblasts upon LPS treatment was shown to be a result of inflammasome priming or activation [8]. We quantified the relative *CTSB* and *CASP1* expression of NHDF treated with LPS to further analyze the impact of LPS stimulation on the mRNA expression of essential inflammasome pathway components. The *CTSB* mRNA expression was slightly induced by LPS in a concentration-dependent manner, while the *CASP1* mRNA expression showed a significant upregulation by 12.1 ± 1.0-fold (*p* < 0.0001) and 9.6 ± 0.7-fold (*p* < 0.0001) in 0.1 and 1.0 μg/mL NHDF treated with LPS, respectively (Figure A1B,C). It can be concluded that LPS is a potent inducer of the gene expression of inflammasome pathway components in nonimmune cells, such as fibroblasts.

### 3.2. Differences in ATP- and LPS-Induced Effects on the XYLT1 mRNA-Expression and XT-I Activity of Primary Skin Fibroblasts

Extracellular ATP, released from dying cells, serves as a damage-associated molecular pattern inducing a pro-inflammatory response in primary fibroblasts [39]; therefore, we investigated ATP-mediated effects on the gene expression profile of NHDF in the presence or absence of LPS. We first examined whether the decreased *XYLT1* and increased *IL1B*, *IL8* mRNA expression mediated by LPS could also be observed upon ATP stimulation of NHDF under high-cell-density-culture conditions for 24 h. The relative mRNA expression of the *XYLT2* isoform was also determined for control purposes (Figure 3).

Similar to the gene expression changes of NHDF treated with LPS mentioned previously (Figure 2 and Figure A1), we observed decreased *XYLT1* mRNA expression and increased *IL1B* mRNA expression (Figure 3A,C), while the *XYLT2* mRNA expression was unchanged in NHDF stimulated with LPS (Figure 3B). By contrast, the ATP treatment of NHDF alone did not alter the expression levels of the latter genes significantly. It is noteworthy that a nonsignificant 0.6 ± 0.1-fold decreased *XYLT1* mRNA expression (*p* = 0.1) was detected upon ATP stimulation (Figure 3A). In addition, the concomitant stimulation of NHDF with LPS and ATP did not modulate the relative *XYLT1* mRNA expression compared to cells solely treated with LPS (Figure 3A). Furthermore, the relative mRNA expression of the *XYLT2* isoform of cells treated simultaneously with LPS and ATP did not differ from that of cells treated solely with LPS (Figure 3B). Compared to NHDF treated with LPS, simultaneous LPS and ATP incubation decreased (0.2 ± 0.0-fold; *p* < 0.0001) the relative *IL1B* transcription of NHDF significantly after a cultivation period of 24 h (Figure 3C). Regarding the gene expression of chemokine *IL8* and the inflammasome components *CTSB* and *CASP1*, no *IL8* expression changes were observed upon ATP treatment alone, while the relative *CTSB* and *CASP1* mRNA expression were slightly increased by 1.2 ± 0.1-fold (*p* < 0.001) and 1.3 ± 0.1-fold (*p* < 0.05), respectively, in cells stimulated with ATP compared to untreated controls. The concomitant stimulation of NHDF with LPS and ATP did not alter the relative mRNA expression of *IL8*, *CTSB* or *CASP1* significantly compared to the respective expression of cells stimulated solely with LPS (Figure A2). Together, these results indicate that ATP treatment alone or in combination with LPS affected the relative gene expression of LPS target genes marginally.

In order to examine whether the LPS-mediated reduction of *XYLT1* mRNA expression correlates with changes in cellular XT-I activity, we determined the extracellular and intracellular XT-I activity of NHDF by UPLC-ESI-MS/MS under comparable experimental conditions to the gene expression analysis (Figure 4).

Compared to control cells, the LPS stimulation at a concentration of 0.1 μg/mL for 24 h did not affect the extracellular XT-I activity, while a slight but not significant decrease in intracellular XT-I activity (0.7 ± 0.1-fold, *p* = 0.4) was observed. The treatment of NHDF solely with ATP diminished the extracellular (0.5 ± 0.1-fold, *p* < 0.001) and intracellular (0.5 ± 0.1-fold, *p* < 0.05) XT-I activity significantly compared to untreated controls after a culture period of 24 h. The concomitant stimulation of NHDF with LPS and ATP resulted in a decreased extracellular (0.4 ± 0.0-fold, *p* < 0.0001) and intracellular (0.6 ± 0.1-fold, *p* < 0.01) XT-I activity compared to NHDF treated solely with LPS for 24 h. No changes in the cellular XT-I activity were detectable in cells treated simultaneously with LPS and ATP and those treated solely with ATP (Figure 4). When the cell treatment period was extended from 24 to 48 h, no significant differences were observed in the extracellular and intracellular XT-I activities of cells treated with LPS compared to untreated cells. The ATP treatment of NHDF for 48 h resulted in a significantly 0.4 ± 0.0-fold decreased extracellular (*p* < 0.0001) and a 0.5 ± 0.0-fold decreased intracellular (*p* < 0.0001) XT-I activity compared to untreated control cells. Furthermore, significant deviations were identified in cells treated simultaneously with LPS and ATP compared to those treated solely with LPS after 48 h since the LPS and ATP treatment resulted in a significantly 0.4 ± 0.0-fold decreased extracellular and intracellular (*p* < 0.0001) XT-I activity. Furthermore, the cellular XT-I activity of NHDF treated concomitantly with LPS and ATP for 48 h did not differ from those of cells treated solely with ATP (Appendix A). We conclude that two distinct regulatory mechanisms exist to control the *XYLT1* expression and the intracellular protein abundance of the cell in the context of inflammatory XT-I regulation.

Previous studies have shown that cell treatment with LPS derived from *Escherichia coli* serotype 0111:B4 is capable of decreasing the fibroblast viability in a concentration- and tissue-specific manner [3,40]. Therefore, we performed a WST-1 reagent-based cell proliferation assay (Figure 5A) to exclude potential proliferation changes of NHDF affecting the relative *XYLT1* mRNA expression decrease by LPS observed in our study.

The proliferation assay showed that neither the concentration of 0.1 nor 1.0 μg/mL LPS affected the fibroblast proliferation and viability after an incubation period of 24 h. Furthermore, neither the single ATP supplementation nor the simultaneous addition of LPS and ATP to the culture medium had an impact on the cell proliferation and viability after 24 h of cell incubation (Figure 5B). It can be assumed that the LPS concentrations applied were well-tolerated by the NHDF for the incubation period of 24 h tested as not affecting the cell viability in our cell culture system.

### 3.3. CASP1 and CTSB Are Negative Regulators of XYLT1 mRNA Expression in Primary Skin Fibroblasts

Having shown that LPS exerts a repressive effect on the *XYLT1* mRNA expression, we wanted to investigate a putative cellular pathway that underlies this regulation. It has been shown previously that the XT-I secretion process was dependent on the cellular activity of cysteine proteases [41], indicating the cysteine proteases CTSB and CASP1 as potential *XYLT1* expression regulators. Based on this assumption, we wanted to evaluate the impact of these inflammasome pathway components on the basal and LPS-regulated *XYLT1* mRNA expression of NHDF. Therefore, we conducted siRNA knockdown experiments in the absence or presence of the *XYLT1* suppressor LPS. The relative gene expression of *CTSB*, *XYLT1* and *CASP1* were quantified 24 h post-transfection with the respective *CTSB* (Figure 6) or *CASP1* (Figure 7) targeting siRNA via qRT-PCR.

A significant 0.01 ± 0.0-fold decreased *CTSB* mRNA expression (*p* < 0.0001) was found in *CTSB*-silenced NHDF in the absence and presence of LPS compared to cells transfected with control siRNA (Figure 6A). Compared to cells transfected with negative control siRNA, the *CTSB*-silenced cells showed a significant 3.4 ± 0.2-fold (*p* < 0.0001) increase in the basal *XYLT1* expression. This relative increase in *XYLT1* expression of *CTSB*-silenced cells remained significantly 2.3 ± 0.1-fold (*p* < 0.0001) higher in the presence of *XYLT1* suppressor LPS than that of control siRNA-treated NHDF (Figure 6B). We determined the relative *CASP1* expression of *CTSB*-silenced and control siRNA-transfected cells to control the specificity of the applied *CTSB* knockdown. Compared to cells treated with non-targeting control siRNA, the *CASP1* mRNA expression decreased 0.4 ± 0.0-fold and 0.6 ± 0.0-fold (*p* < 0.0001) in *CTSB*-silenced NHDF that were cultured in the absence and presence of LPS (Figure 6C). These results demonstrate that CTSB might be a negative regulator of *XYLT1* mRNA expression under physiological and inflammatory conditions. Since a simultaneous decrease in the relative *CASP1* expression occurred in *CTSB*-silenced cells, it cannot be ruled out that the increased *XYLT1* expression observed is mediated solely by *CTSB* suppression.

We next performed a siRNA-mediated knockdown of *CASP1* in NHDF and quantified the relative gene expression of *CASP1, XYLT1* and *CTSB* 24 h post-transfection (Figure 7).

We observed a significant 0.1 ± 0.0-fold (*p* < 0.0001) decreased *CASP1* mRNA expression in *CASP1*-silenced NHDF in the absence and presence of LPS compared to cells transfected with control siRNA (Figure 7A). The *CASP1*-silenced cells revealed a significant 2.4 ± 0.2-fold (*p* < 0.0001) increase in the basal *XYLT1* expression compared to NHDF transfected with negative control siRNA. The *XYLT1* expression of *CASP1*-silenced cells remained significantly 2.0 ± 0.2-fold (*p* < 0.0001) increased in the presence of LPS compared to cells treated with control siRNA (Figure 7B). We determined the relative *CTSB* mRNA expression of *CASP1*-silenced cells to detect any potential regulatory loops between *CASP1* and *CTSB*. No significant differences in the relative *CTSB* mRNA expression were observed between *CASP1*-silenced and control siRNA-transfected NHDF. A slight increase in *CTSB* expression (1.3 ± 0.1-fold (*p* < 0.001)) was detectable in *CASP1*-silenced cells in the presence of LPS compared to control siRNA treatment (Figure 7C). These results show that the cysteine protease CASP1 is a negative regulator of the *XYLT1* mRNA expression of NHDF under physiological and inflammatory conditions mimicked by the absence and presence of LPS in our cell culture system.

### 3.4. PXE Fibroblasts Exhibit a Nonsignificant Reduction in XYLT1 mRNA Expression

The inherited metabolic disease PXE has previously been shown to involve aberrant gene expressions associated with the inflammatory IL-1β pathway [29]. After showing that inflammatory pathway components are negative regulators of *XYLT1* mRNA expression, we used PXE fibroblasts and a cell culture model established formerly [28] mimicking the disease conditions of PXE to independently confirm the gene expression patterns observed in this study. Since the PXE cell culture model uses LPDS instead of FCS supplementation of the growth medium, we had to confirm that the LPS-mediated effects on the *XYLT1* mRNA expression observed above were reproducible under the high-cell-density-culture condition with LPDS. The quantified gene expression changes of *XYLT1*, *XYLT2* and *IL1B* in NHDF stimulated with LPS cultured in the presence of LPDS resemble those of NHDF that were maintained in FCS-containing medium (Figure A3).

After verifying that the PXE cell culture model with LPDS is suitable for further gene expression analyses, we cultured NHDF and PXE fibroblasts and compared their cellular response upon LPS treatment. The relative gene expression of *XYLT1*, *CTSB* and *CASP1,* and that of the LPS-inducible genes *IL1B* and *IL8,* was analyzed by qRT-PCR (Figure 8).

Compared to NHDF, we observed a slight but not significant decrease in the relative *XYLT1* mRNA expression of untreated PXE fibroblasts (0.6 ± 0.1-fold; *p* = 0.4). The LPS stimulation resulted in both NHDF and PXE fibroblasts in 0.3 ± 0.1-fold and 0.4 ± 0.0-fold decreased *XYLT1* mRNA expression (*p* < 0.0001), respectively, compared to the corresponding untreated control cells. Therefore, the *XYLT1* mRNA expression of NHDF treated with LPS and PXE fibroblasts treated with LPS did not differ from each other (Figure 8A). Regarding the basal *CTSB* mRNA expression of PXE fibroblasts, a significant 2.3 ± 0.2-fold increased expression level (*p* < 0.0001) was detected compared to NHDF. The LPS treatment of NHDF did not change the relative *CTSB* mRNA expression compared to untreated NHDF, while the LPS treatment of PXE fibroblasts resulted in a slight 1.3 ± 0.2-fold increased expression level (*p* < 0.05) compared to untreated PXE fibroblasts. Therefore, a 2.9 ± 0.3-fold higher *CTSB* expression level was detected in PXE fibroblasts treated with LPS relative to NHDF treated with LPS (Figure 8B). The basal *CASP1* mRNA expression of PXE fibroblasts did not differ from that of NHDF. In comparison to untreated control cells, the LPS stimulation of NHDF or PXE fibroblasts revealed a significant 12.6 ± 0.8-fold or 17.5 ± 1.7-fold (*p* < 0.0001) increase in the *CASP1* expression, respectively. Thus, a significant 1.4 ± 0.1-fold higher *CASP1* expression level was detected in PXE fibroblasts treated with LPS relative to NHDF treated with LPS (Figure 8C). The basal expression of cytokine *IL1B* did not differ between PXE fibroblasts and NHDF. The LPS treatment of NHDF resulted in a 505 ± 81-fold (*p* < 0.0001) increased *IL1B* mRNA expression compared to untreated controls, while PXE fibroblasts showed a 1932 ± 320-fold (*p* < 0.0001) increased *IL1B* expression level in comparison with untreated PXE cells. Therefore, the relative *IL1B* expression of PXE fibroblasts treated with LPS was 2.8 ± 0.4-fold (*p* < 0.0001) higher than that of NHDF treated with LPS (Figure 8D). The basal expression of chemokine *IL8* did not differ between PXE fibroblasts and NHDF either. The LPS treatment resulted in a significant 768 ± 145-fold increased *IL8* expression in NHDF and a 2088 ± 188-fold (*p* < 0.0001) increased in PXE fibroblasts compared to the respective untreated control cells. The *IL8* expression level of PXE fibroblasts treated with LPS was 2.8 ± 0.4-fold (*p* < 0.0001) higher than that of NHDF treated with LPS (Figure 8E). We conclude from the results that PXE fibroblasts possess higher sensitivity towards exogenous LPS, resulting in a more pronounced inflammatory gene expression change induced by LPS compared to NHDF. Furthermore, our data provide, for the first time, a correlation of decreased *XYLT1* expression with the previously observed decrease in basal XT activity of PXE fibroblasts compared to NHDF that might involve basal expression differences of inflammatory pathway components, such as *CTSB.*

## 4. Discussion

Despite playing an essential role in ECM remodeling, fibroblasts are also important key sentinel cells that activate and modulate immune responses upon the recognition of pathological stimuli [38]. The identification of disease-specific alterations in fibroblasts is the crucial step in unraveling the molecular pathways underlying pathological changes. Our in vitro cell culture model used LPS and ATP to simulate an inflammatory environment. As shown by the WST-1 cell proliferation assay, both LPS concentrations of 0.1 and 1.0 μg/mL, as well as sole ATP treatment or ATP in combination with LPS, did not significantly affect the number of metabolically active primary fibroblasts after 24 h of cell treatment. Thus, it can be concluded that fibroblasts are quite resistant to the noxious stimuli that have previously been shown to induce apoptosis in other cell types [42,43]. Consistent with our data, the resistance of human fibroblasts towards exogenous LPS was demonstrated in numerous studies utilizing LPS concentrations ranging up to 10 μg/mL and incubation periods of 48 to 72 h [4,6,38]. This highlights the key role of fibroblasts as immune sentinel cells affecting both the inflammatory and repair processes during wound healing and tissue homoeostasis.

Fibroblasts actively define the structure of tissue microenvironments and regulate inflammatory responses by the production of cytokines and chemokines, such as IL-1β and IL-8 [44]; therefore, we analyzed the relative expression of *IL1B* and *IL8* upon the LPS treatment of NHDF as additional inflammatory markers. In agreement with the literature [4,38,45], NHDF showed a high reaction towards LPS, resulting in increased gene expression levels of both markers. These data further strengthened the view of fibroblasts as key immune sentinel cells sensing pathogenic LPS, on the one hand, and recruiting leukocytes by the expression of chemokines, such as IL-8, on the other, contributing to the initiation of the inflammatory response during tissue damage.

Elevated levels of CTSB in fibroblasts are observed in inflammatory diseases, including rheumatoid arthritis [13,46]. Regarding immune responses and inflammation, CTSB plays an important role in both the NF-κB and CASP1 activation [15,16,47]. In the present study, we determined that LPS increased the *CTSB* mRNA expression in NHDF, similar to previous findings in human fibroblast cell lines [13]. In agreement with data in human fibroblasts [7], we observed the LPS-induced upregulation of *CASP1* mRNA expression in NHDF. We also found that the ATP treatment alone induced both *CTSB* and *CASP1* mRNA expression in NHDF. This result is in line with the data on human fibroblasts showing that ATP treatment alone induces CASP1 activation [7]. It can be concluded that the LPS and ATP treatment used in our NHDF model system is sufficient to simulate an inflammatory microenvironment in vitro.

The gene expression and activity of the GAG-initiating key enzyme XT-I was found to be differentially regulated during disease conditions. Up to now, studies have described an induction of *XYLT1* expression in fibrotic tissues or human primary fibroblasts mediated by cytokines [5,21,48,49], while, to the best of our knowledge, the suppression of its mRNA expression and activity has not been described previously in the context of fibroblast-mediated inflammatory responses. In the present study, we demonstrated for the first time that LPS has a regulatory effect on the relative *XYLT1* mRNA expression of NHDF that is independent of the cell culture density and type of FCS used. The *XYLT1* mRNA expression decrease observed in both low- and high-cell-density-culture models is consistent with previous cross-tissue examinations of fibroblasts that highlighted shared fibroblast phenotypes across a spectrum of inflammatory and fibrotic diseases [50,51]. The activated NF-kB in fibroblasts stimulated with LPS was shown to induce Smad7 gene expression [27]. Since the regulation of *XYLT1* mRNA expression is mediated via the MAPK and Smad pathway in NHDF [34], it can be assumed that LPS decreases the relative *XYLT1* mRNA expression by inducing the *SMAD7* mRNA expression. This hypothesis is further strengthened by the fact that in the presence of growth factors, *SMAD7*-targeting siRNA-transfected cells show a more pronounced *XYLT1* mRNA expression increase compared to NHDF transfected with control siRNA [34]. In contrast to the LPS-mediated *XYLT1* expression decrease, marginal or no changes of the relative *XYLT2* mRNA expression were observed upon the LPS treatment of NHDF. This difference in the *XYLT1* and *XYLT2* isoform expression was also observed in cytokine-stimulated NHDF [5,34] and was assumed to be based on differences in the transcriptional regulation of the corresponding promotor regions [52]. Future studies will be necessary to evaluate the differences in *XYLT1* and *XYLT2* expression mediated by LPS and analyze the involvement of the *XYLT2* isoform in immunoregulatory processes.

Articular cartilage damage is a key event leading to joint deformity in rheumatoid arthritis, osteoarthritis, and septic arthritis. The expression of GAG-initiating key enzyme XT-I was found to be downregulated in human osteoarthritis and diminished by an amino-terminal fibronectin fragment, a damage-associated molecular pattern, in chondrocytes [23]. In agreement with previous reports, our data showed that inflammatory LPS stimulation reduced the relative *XYLT1* mRNA expression of NHDF. Interestingly, the decrease in *XYLT1* mRNA expression observed in this study at 24 h post LPS treatment was not visible on the enzyme activity level at 24 or 48 h. Since previous studies have shown that an increase in cellular XT-I activity results from a former time-dependent change in the *XYLT1* mRNA expression of NHDF [34], these results provide a strong argument for the existence of regulatory and kinetic differences between XT-I synthesis and XT-I turnover. This assumption is further strengthened by our finding that the ATP treatment of NHDF did not change the relative *XYLT1* expression of NHDF significantly but reduced the intracellular XT-I activity significantly at 24 h compared to untreated cells. Whether the relative reduction in intracellular XT-I activity results from the increased shedding and secretion of the XT-I into the extracellular space was not clarified in this study. Previous studies have shown the inhibitory potential of nucleotides on the in-silico measurement of the XT activity [53]. Despite showing a decrease in extracellular XT-I activity by ATP in our cell culture model, we assume a false negative result due to the presence of the ATP in the cell culture supernatant. This limitation can be overcome in future investigations by using a different experimental setup, such as including a medium change and further incubation of the cells in ATP-free media for additional 48 h. In conclusion, these results provide new evidence for mechanistic differences in inflammation-mediated XT-I suppression and fibrotic XT-I induction in NHDF.

Despite the fact that CTSB has been shown to degrade collagens in fibroblasts [13], its role in regulating *XYLT1* mRNA expression by fibroblasts during chronic inflammation has not been analyzed before. We found here, for the first time, by performing siRNA-mediated gene knockdown experiments, that the two inflammasome components (CTSB and CASP1) were negative regulators of the *XYLT1* mRNA expression in NHDF. However, we were unable to differentiate between solely CTSB- and CASP1-mediated effects on the *XYLT1* expression of NHDF. Although siRNA-mediated *CASP1* knockdown did not affect the expression of *CTSB*, indicating CASP1 as a potent *XYLT1* expression inhibitor, the siRNA-mediated *CTSB* knockdown resulted in a simultaneous reduction of basal *CASP1* expression. These data indicate a critical role for CTSB in CASP1-mediated effects. This hypothesis is supported by the finding that CTSB and CASP1 were colocalized and that treatment with a CTSB inhibitor markedly inhibited CASP1 expression in mouse models of inflammatory pain [54,55,56]. In summary, we have elucidated a novel mechanism for CTSB and CASP1 to alleviate the decreased *XYLT1* expression in inflammatory conditions (Figure 9). Due to the correlation of *XYLT1* mRNA expression and XT-I enzyme activity increase shown in numerous other studies using human dermal and cardiac fibroblasts [21,34,48], we presume that the relative *XYLT1* expression increase in CTSB and CASP1 knockdown cells will result in higher cellular XT-I activities. This assumption is supported by a previous study that showed the involvement of cysteine proteases in the secretion process of XT-I. As the cysteine protease inhibitor cocktail used in the study mentioned previously contained inhibitors with specificity for the cathepsins B, L and S, and for proteasomes and papain [41], our investigation here pointed towards the role of CTSB in cellular XT-I regulation. Future studies to verify these initial results should evaluate the enzyme activity of the knockdown cells and include the use of specific small molecule inhibitors targeting inflammasome components.

Identifying factors regulating the fibroblast phenotype in one disease may be repurposed to treat other diseases. Fibroblasts in PXE are characterized by the expression of NF-κB downstream targets such as IL-6 and increased expression of genes directly associated with cholesterol biosynthesis [28]. Furthermore, a decreased XT activity was shown in primary PXE fibroblasts compared to NHDF under low cell density culture conditions [57]. The XT activity was reported to decrease with the cartilage age in rats [58], while the activity of CTSB increases significantly with age [59], and the involvement of this enzyme in inflammation and cholesterol trafficking in macrophages has been demonstrated [33]. Therefore, we assume that the aberrant expression of inflammatory pathway components might contribute to the characteristics of PXE fibroblasts described above. Consistent with this hypothesis, we found LPS to induce a more pronounced inflammatory response, indicated by a higher increase in the relative expression of inflammatory genes, including *CTSB*, *CASP1*, *IL1B* and *IL8*, in PXE fibroblasts compared to NHDF. Based on the data from this study, we present new evidence that PXE involves an aberrant gene expression of inflammatory pathway components in fibroblasts. Using the *XYLT1* expression as a marker for inflammatory pathway involvement in primary fibroblasts, we found a slight but not significant basal *XYLT1* expression decrease in PXE fibroblasts compared to NHDF. It is noteworthy that PXE fibroblasts possessed a significantly higher basal *CTSB* mRNA expression than NHDF, which correlates reciprocally with the reduced basal *XYLT1* mRNA expression. This finding resembles those of previous works showing lower XT activity in PXE fibroblasts or aberrant gene expressions in PXE that are associated with the inflammatory IL-1β pathway [29,57]. We conclude that the decreased *XYLT1* mRNA expression can be utilized as a predictive tool for the involvement of inflammatory responses within primary fibroblasts, independently of the cell density and culture medium supplementation used.

Manipulating inflammatory signaling in fibroblasts may lead to novel treatment strategies for inflammatory diseases. Therefore, future studies should address the XT-I and CTSB enzyme activity as potential treatment approaches in the context of inflammatory diseases such as PXE.

## Figures and Tables

**Figure 1 biomedicines-10-01451-f001:**
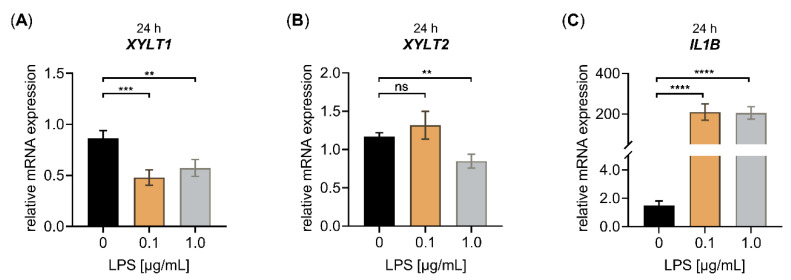
Effect of LPS on the relative *XYLT1*, *XYLT2* and *IL1B* mRNA expression of primary fibroblasts cultured in low cell density culture conditions for 24 h. The NHDF (*n* = 3) were cultured at a density of 50 cells/mm^2^ in DMEM with 10% (*v*/*v*) FCS for 24 h. Treatment was performed with either 0 μg/mL LPS (black), 0.1 μg/mL LPS (orange) or 1.0 μg/mL LPS (grey) supplemented growth medium. The relative expression of (**A**) *XYLT1*, (**B**) *XYLT2* and (**C**) *IL1B* was determined after the LPS treatment of NHDF for 24 h by qRT-PCR. All data are presented as means ± SEM of biological and technical triplicates per donor-derived primary cell culture. Statistical analysis was performed by Mann–Whitney *U* test: ns (not significant), ** *p* < 0.01, *** *p* < 0.001 and **** *p* < 0.0001.

**Figure 2 biomedicines-10-01451-f002:**
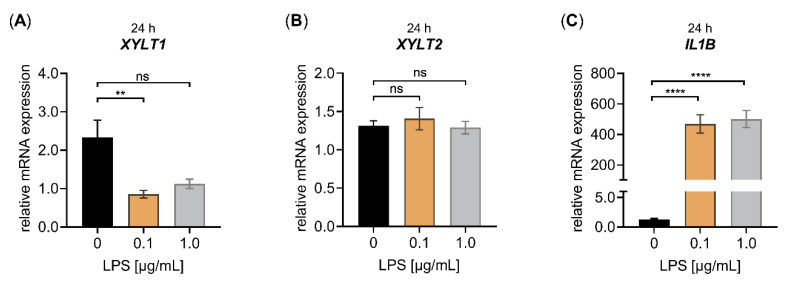
Effect of LPS on the relative *XYLT1*, *XYLT2* and *IL1B* mRNA expression of primary fibroblasts cultured under high-cell-density-culture conditions for 24 h. The NHDF (*n* = 3) were cultured at a density of 177 cells/mm^2^ in DMEM with 10% (*v*/*v*) FCS for 24 h. Treatment was performed with either 0 μg/mL LPS (black), 0.1 μg/mL LPS (orange) or 1.0 μg/mL LPS (grey) supplemented growth medium. The relative expression of (**A**) *XYLT1*, (**B**) *XYLT2* and (**C**) *IL1B* was determined after the LPS treatment of NHDF for 24 h by qRT-PCR. All data are presented as means ± SEM of biological and technical triplicates per donor-derived primary cell culture. Statistical analysis was performed by Mann–Whitney *U* test: ns (not significant), ** *p* < 0.01 and **** *p* < 0.0001.

**Figure 3 biomedicines-10-01451-f003:**
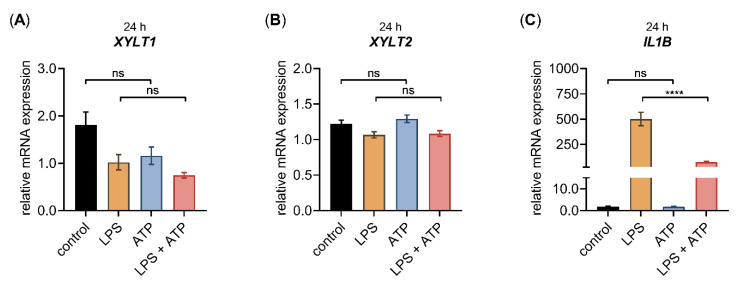
The relative *XYLT1*, *XYLT2* and *IL1B* mRNA expression of primary fibroblasts after LPS and ATP treatment for 24 h. The NHDF (*n* = 3) were cultured at a cell density of 177 cells/mm^2^ in growth medium supplemented with 10% (*v*/*v*) FCS for 24 h. Cells were treated with either 0.1 μg/mL LPS (orange), 5 mM ATP (blue) or both 0.1 μg/mL LPS, and 5 mM ATP (red), for 24 h. The relative expression of (**A**) *XYLT1*, (**B**) *XYLT2* and (**C**) *IL1B* of NHDF was determined by qRT-PCR. All data are presented as means ± SEM of biological and technical triplicates per donor-derived primary cell culture. Statistical analysis was performed by Mann–Whitney *U* test: ns (not significant), **** *p* < 0.0001.

**Figure 4 biomedicines-10-01451-f004:**
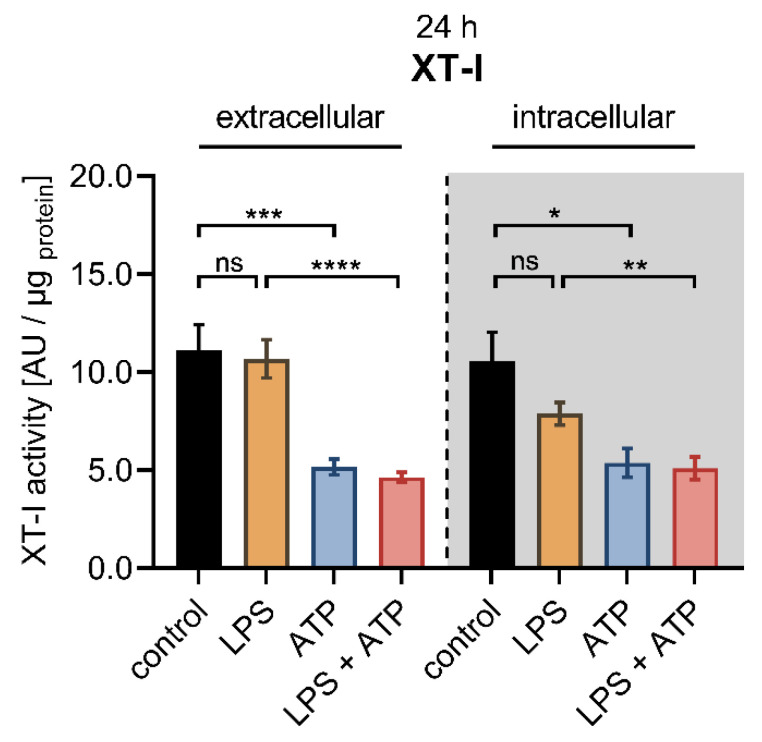
Cellular XT-I activity of primary fibroblasts after LPS and ATP treatment. The NHDF (*n* = 3) were cultured at a cell density of 177 cells/mm^2^ in DMEM supplemented with 10% (*v*/*v*) FCS for 24 h. Cells were treated with either 0.1 μg/mL LPS (orange), 5 mM ATP (blue) or both 0.1 μg/mL LPS and 5 mM ATP (red) for 24 h. The cellular XT-I activity was determined in the cell culture supernatants (extracellular) and the cell lysates (intracellular, grey shaded) by UPLC-ESI-MS/MS assay. The intracellular and extracellular XT-I activity is given in arbitrary units (AU) per μg of protein. All data are presented as means ± SEM of biological and technical triplicates per donor-derived primary cell culture. Statistical analysis was performed by Mann–Whitney *U* test: ns (not significant), * *p* < 0.05, ** *p* < 0.01, *** *p* < 0.001 and **** *p* < 0.0001.

**Figure 5 biomedicines-10-01451-f005:**
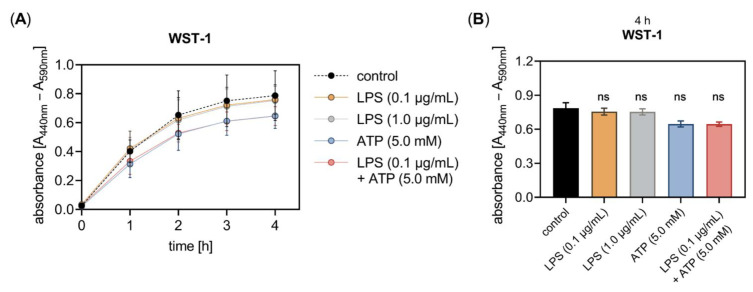
The LPS and ATP treatment did not affect the in vitro cell proliferation of primary fibroblasts. The NHDF (*n* = 3) were cultured on 96-well microtiter plates for 24 h. (**A**) Cells were treated with either 0.1 μg/mL LPS (orange), 1.0 μg/mL LPS (grey), 5 mM ATP (blue), or both 0.1 μg/mL LPS and 5 mM ATP (red) for 24 h. The cell proliferation was determined 0, 1, 2, 3 and 4 h after WST-1 reagent supplementation at the 20 h time point. The absorbance was measured at 440 nm, with 590 nm as a reference wavelength. (**B**) The WST-1 assay results measured 4 h after the WST-1 supplementation are shown as bar graphs. All data are presented as means ± SEM of five biological replicates and one technical replicate per donor-derived cell culture. Statistical analysis was performed by Mann–Whitney *U* test: ns (not significant) when compared to the untreated cells (control, black).

**Figure 6 biomedicines-10-01451-f006:**
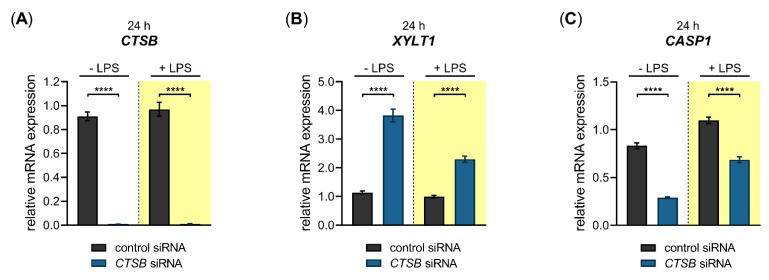
Basal and LPS-regulated *XYLT1* mRNA expression after siRNA-mediated *CTSB* knockdown in primary fibroblasts. The NHDF (*n* = 3) were treated with a non-targeting control siRNA (black) or a targeting siRNA against *CTSB* (blue); 24 h post-transfection, cells were maintained in growth medium supplemented without or with 0.1 μg/mL LPS (highlighted yellow) for an additional 24 h. The relative expression of (**A**) *CTSB*, (**B**) *XYLT1* and (**C**) *CASP1* was determined by qRT-PCR. All data are presented as means ± SEM of biological and technical triplicates per donor-derived cell culture. The dashed lines indicate that both experiments were performed independently and, therefore, the relative gene expression values were related to the respective cell treatments with control siRNA. Statistical analysis was performed by Mann–Whitney *U* test: **** *p* < 0.0001.

**Figure 7 biomedicines-10-01451-f007:**
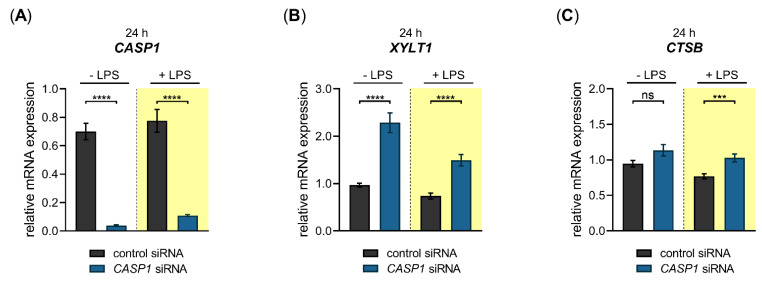
Basal- and LPS-regulated *XYLT1* mRNA expression after siRNA-mediated *CASP1* knockdown in primary fibroblasts. The NHDF (*n* = 3) were treated with either a non-targeting control siRNA (black) or a targeting siRNA against *CASP1* (blue); 24 h post-transfection, cells were maintained in growth medium supplemented without or with 0.1 μg/mL LPS (highlighted yellow) for an additional 24 h. The relative expression of (**A**) *CASP1*, (**B**) *XYLT1* and (**C**) *CTSB* was determined by qRT-PCR. All data are presented as means ± SEM of biological and technical triplicates per donor-derived primary cell culture. The dashed lines indicate that both experiments were performed independently and, therefore, the relative gene expression values were related to the respective cell treatments with control siRNA. Statistical analysis was performed by Mann–Whitney *U* test: ns (not significant), *** *p* < 0.001 and **** *p* < 0.0001.

**Figure 8 biomedicines-10-01451-f008:**
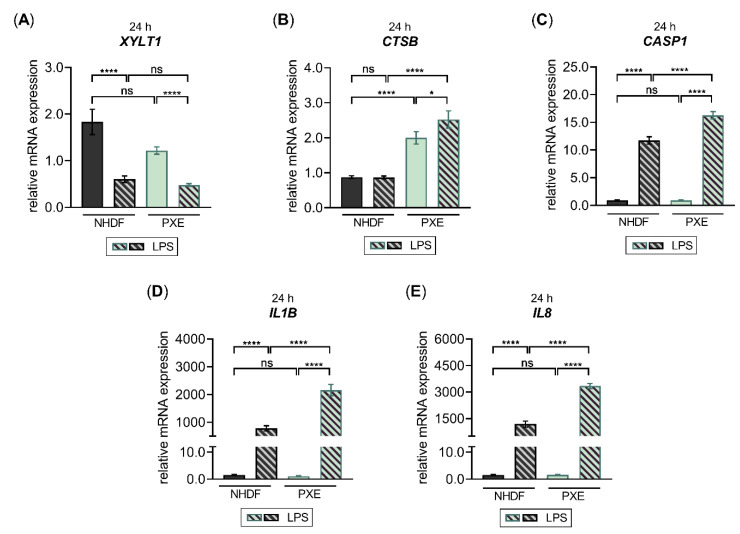
The LPS-regulated gene expression of *XYLT1* and inflammatory pathway components in PXE fibroblasts. The NHDF (*n* = 3) were maintained at a cell density of 177 cells/mm^2^ in DMEM with 10% (*v*/*v*) LPDS for 24 h. The treatment was performed either without or with 0.1 μg/mL LPS for 24 h. The relative expression of (**A**) *XYLT1*, (**B**) *CTSB*, (**C**) *CASP1*, and (**D**) *IL1B* and (**E**) *IL8* was determined by qRT-PCR. All data are presented as means ± SEM of biological and technical triplicates per donor-derived primary cell culture. Statistical analysis was performed by Mann–Whitney *U* test: ns (not significant), * *p* < 0.05, **** *p* < 0.0001.

**Figure 9 biomedicines-10-01451-f009:**
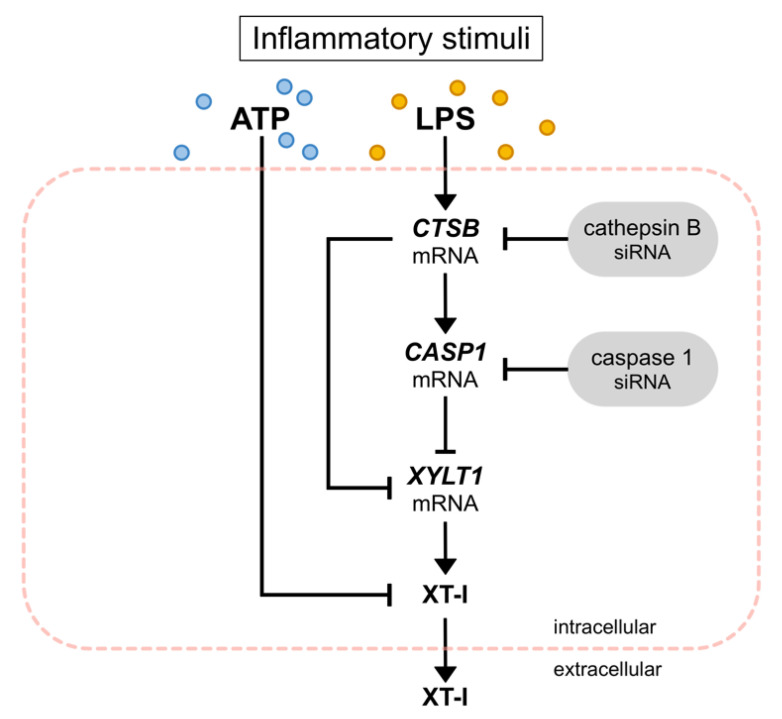
Inhibition of *XYLT1* expression and XT-I activity by inflammatory stimuli. Exogenous ATP reduced the intracellular XT-I activity of NHDF. LPS increased the mRNA expression of *CTSB* and *CASP1* while decreasing the expression of *XYLT1*. Performing siRNA-mediated knockdown experiments, CTSB and CASP1 were identified as negative regulators of *XYLT1* mRNA expression.

## Data Availability

Not applicable.

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
