# Peer review of "The Impact of Inflammatory Stimuli on Xylosyltransferase-I Regulation in Primary Human Dermal Fibroblasts"

_biomedicines, 2022, doi:10.3390/biomedicines10061451_

Round 1
Reviewer 1 Report
Ly et al, show the effects of LPS stimulation on the GAG biosythesis enzyme XT in fibroblasts. They show in detail that LPS suppresses XYLT1 gene expression through Caspase 1 via siRNA knockdown. Finally they confirm their work in a disease setting ie PXE which has abberent IL-1B activity (which is downstream of LPS stimulation). The paper is well written and the conclusion drawn from the results are sound. I have a few suggestion.
1. In figure 1 the authors show LPS reduces XYLT1 expression. As XYTL1 is pro-fibrotic this suggests LPS is having antifibrotic effects. The authors cite ref 21 to describe the effects of LPS on pro-fibrotic gene expression but this ref does not include effects of LPS on pro-fibrotic gene expression. Could the authors cite evidence of the effects of LPS on pro-fibrotic gene expression or better still could the authors look at alpha sma or fibronectin expression in the samples from figure 1.
2. The authors show Caspase 1 KO enhances XYTL1 mRNA expression but do not look at the effects of caspase 1 KO on XT-1 activity as shown in figure 4. Could the authors perform the assay described in figure 4 on caspase 1 KO cells to determine if the increase in XYLT1 mRNA translates to increased XT-1 activity.
3. I believe the readers would be aided with a schematic at the end describing the role LPS plays in suppressing XYLT1.
Reviewer 2 Report
The title of the article fully reflects the content of the article.
The Abstract section contains the necessary information for the reader: for example, background information on silosyltransferase-I (XT-I) and its role in the biosynthesis of proteoglycans. The relevance of the study is presented, which is expressed in the lack of understanding of the role of inflammation in the regulation of XT-I. The abstract outlines the purpose of the study, methods and main results of the study. The conclusions are clear and consistent with the results of the study. The information presented in the article improves the mechanistic understanding of the inflammatory regulation of XT-I. It can help in the development of strategies for therapy aimed at fibroblasts in inflammatory diseases.
The presented keywords are necessary and reflect the research topic presented by the authors.
In the Introduction section, the authors presented a general understanding of what constitutes inflammation and what place it occupies in the development of inflammatory and autoimmune diseases, as well as in the regeneration of damaged tissue. The Introduction notes the important role of tissue-resident fibroblasts in the synthesis of extracellular matrix (ECM), remodeling and maintenance of tissue homeostasis, activation and modulation of immune responses. The mechanism of regulation of inflammation by fibroblasts is singled out separately. A significant part of the "Introduction" is devoted to silosyltransferase-I, proteoglycans and IL-1ß. The purpose of the study is clear. The connection between the articles cited in the "Introduction" and the purpose of this study is visible.
In the section "Materials and methods", the preparation of materials and reagents for the study is presented quite fully and correctly. The cells (NSDF and dermal fibroblasts from patients with PXE), their processing and preparation of samples for the study are fully described. The section presents the analysis of cell proliferation, bicynchonicic acid, determination of the activity of XT-I, methods of RNA extraction and complementary DNA synthesis, and quantitative real-time PCR analysis. Table 1 in the necessary section. At the conclusion of the section, the authors presented a statistical analysis of the results of the study.
Please note that the section "Materials and methods" does not provide the approval of the ethics committee for the study. This approval must be submitted.
All the information in the section "Materials and Methods" is extensive and necessary. The design of the study is clear. In the "Methods" section, the authors make references to previously conducted work, which is necessary to understand the researchers' intention.
The "Results" section presents the main results of the study, while they are illustrated with figures. All drawings are legible, necessary and complement the content of the article. All the tasks planned by the authors have been completed. The results presented in the article are necessary because they lay the foundation for the development of approaches to the treatment of inflammatory diseases.
In the "Discussion" section, using the literature, the authors discussed the research results. The article contains certain limitations: all studies were performed in vitro. In the future, it is necessary to confirm the results obtained in experiments on animal models of inflammation and human cells.
The correct conclusions follow from the results of the conducted research.
The submitted manuscript does not cause any concerns. The manuscript does not cause any ethical problems, except for the lack of permission from the ethics committee to conduct research. It is necessary to supplement the section "Materials and methods" with the permission of the ethics committee to conduct a study. All references to publications presented by the authors in the article are necessary and correct, made in the right style. Out of 70 links to publications, 24 have been published over the past 5 years.
I have no concerns about the similarity of this article with other articles published by the same authors.
Round 2
Reviewer 1 Report
The authors have not included a rebuttal to the reviewers comments in the re-submission. From the marked manuscript provided by the authors I can see that the authors have not addressed any of the comments from the first review.
In line 303, the authors state that the impact of LPS on pro-fibrotic gene expression is described in ref 21. It was highlight in the first review that ref 21 includes no data showing the effects of LPS in pro-fibrotic gene expression. Could the authors provide evidence in the literature of the pro-fibrotic role of LPS.
In the first review it was suggested that the authors provide a schematic at the end of the manuscript to summary the role LPS plays in suppressing XYLT1. I still believe a schematic would be beneficial to the readers.
Author Response
The word document of the point-by-point response was not successfully uploaded last time. Please accept our apology for the inconvenience caused. Please see the attachment for the last point-by-point response provided by us.

Round 3
Reviewer 1 Report
The authors have clarified in the rebuttal the issues I raised in review. If the authors include the schematic (shown in the rebuttal) in the main text then I believe the manuscript is now suitable for publication.
